

# Advances in global sensitivity analyses of demographic-based species distribution models to address uncertainties in dynamic landscapes

Ilona Naujokaitis-Lewis[1,2] and Janelle M.R. Curtis[2,3]

[1] National Wildlife Research Centre, Carleton University, Environment and Climate Change Canada, Ottawa, Ontario, Canada
[2] Centre for Applied Conservation Research, University of British Columbia, Vancouver, British Columbia, Canada
[3] Conservation Biology Section, Pacific Biological Station, Fisheries and Oceans Canada, Nanaimo, British Columbia, Canada

Corresponding author
Ilona Naujokaitis-Lewis,
ilona.naujo.lewis@gmail.com,
ilona.naujokaitis-lewis@canada.ca

## ABSTRACT

Developing a rigorous understanding of multiple global threats to species persistence requires the use of integrated modeling methods that capture processes which influence species distributions. Species distribution models (SDMs) coupled with population dynamics models can incorporate relationships between changing environments and demographics and are increasingly used to quantify relative extinction risks associated with climate and land-use changes. Despite their appeal, uncertainties associated with complex models can undermine their usefulness for advancing predictive ecology and informing conservation management decisions. We developed a computationally-efficient and freely available tool (GRIP 2.0) that implements and automates a global sensitivity analysis of coupled SDM-population dynamics models for comparing the relative influence of demographic parameters and habitat attributes on predicted extinction risk. Advances over previous global sensitivity analyses include the ability to vary habitat suitability across gradients, as well as habitat amount and configuration of spatially-explicit suitability maps of real and simulated landscapes. Using GRIP 2.0, we carried out a multi-model global sensitivity analysis of a coupled SDM-population dynamics model of whitebark pine (*Pinus albicaulis*) in Mount Rainier National Park as a case study and quantified the relative influence of input parameters and their interactions on model predictions. Our results differed from the one-at-time analyses used in the original study, and we found that the most influential parameters included the total amount of suitable habitat within the landscape, survival rates, and effects of a prevalent disease, white pine blister rust. Strong interactions between habitat amount and survival rates of older trees suggests the importance of habitat in mediating the negative influences of white pine blister rust. Our results underscore the importance of considering habitat attributes along with demographic parameters in sensitivity routines. GRIP 2.0 is an important decision-support tool that can be used to prioritize research, identify habitat-based thresholds and management intervention points to improve probability of species persistence, and evaluate trade-offs of alternative management options.

## INTRODUCTION

Addressing the realities and complexities of population-level responses to global change increasingly requires the integration of multiple modeling approaches that capture changes in habitat and climatic suitability and population dynamics, while accounting for the relationship between the environment and demographics (*Schurr et al.*, *2012*). Population dynamics models coupled with species distribution models (SDMs; *Guisan & Thuiller*, *2005*) is one modeling framework that is used increasingly to quantify relative risks of extinction, and guide scientific research and management decisions for rare or threatened species (*Keith et al.*, *2008*; *Anderson et al.*, *2009*; *Naujokaitis-Lewis et al.*, *2013*). By capturing many of the key ecological processes and mechanisms, coupled SDM-population dynamics models represent a powerful simulation-based approach to predict the consequences of populations to dynamic future scenarios of land-use and climate change (*Franklin*, *2010*). Additionally, the explicit modeling of demographic processes that directly relate to extinction risk addresses some of the challenges with interpretation of SDM outcomes, where changes in area of suitable habitat are often simply assumed to be linearly related to extinction risk (*Thuiller et al.*, *2008*; *Fordham et al.*, *2012*).

While the increasing realism of coupled SDM-population dynamics models is an attractive feature, the reliance on extensive ecological and landscape data and multi-model integrations results in complex models with many potential types and sources of uncertainty. Of particular relevance, epistemic uncertainties, which include uncertainty related to natural variation of ecological systems, model uncertainties (i.e., model specification), and parameter uncertainties (*Elith, Burgman & Regan*, *2002*; *Regan, Colyvan & Burgman*, *2002*), are associated with each of the sub-models, as well as the functions that link them. For example, in the context of SDMs, various factors can affect model predictive power and the resultant spatial prediction outcomes, including the type of SDM algorithm and variables selected to model habitat suitability (*Barry & Elith*, *2006*; *Dormann et al.*, *2008*), while the choice of thresholds to distinguish suitable habitat from unsuitable areas can introduce further errors into maps (*Liu, White & Newell*, *2011*). When the habitat map derived from the SDM forms the basis for defining the spatial structure of a (meta-) population (i.e., number, size, shape, and location of patches or populations), demographic parameters may be expressed as functions of map-based habitat attributes. Subsequently, uncertainties associated with SDMs can propagate through to predictions of (meta-) population trajectories and metrics including estimates of extinction risk. The predictions from coupled SDM-population dynamics models are also influenced by uncertainties associated with the specification of relationships and parameters used to model demographic processes. Population models are often formulated with complex relationships, which can introduce substantive uncertainties in expectations due to a

combination of model structure and parameter uncertainties (*Reed et al.*, *2002*; *Naujokaitis-Lewis et al.*, *2009*; *Zurell et al.*, *2011*).

Despite the advantages of coupled SDM-population dynamics models, it is important to understand the potential propagation and influence of uncertainties resulting from each of the sub-models to avoid a false sense of confidence in outcomes and identify management options that are robust to those uncertainties. Sensitivity analyses are used to identify factors that influence uncertainty in the predictions, identify research priorities for reducing uncertainty, evaluate competing model structures, and compare the expected performance of alternative management scenarios (*Cross & Beissinger*, *2001*; *Saltelli et al.*, *2006*). While extensive recent efforts address uncertainties in SDMs when used independently (e.g., *Dormann et al.*, *2008*; *Buisson et al.*, *2010*), evaluating uncertainties and their propagation through to coupled SDM-population dynamics model predictions is not consistently considered (*Cabral et al.*, *2011*; *Zurell et al.*, *2011*; *Prowse et al.*, *2016*). Even the most commonly varied parameters in sensitivity analyses of coupled SDM-population dynamics models, namely fecundities and survival rates, are inconsistently assessed for their influence on model predictions (*Naujokaitis-Lewis et al.*, *2009*). Performing sensitivity analyses that address the contribution of uncertainty across all sub-models of a coupled or integrated model is an important step prior to applying complex ecological models to address conservation problems.

Consideration of uncertainty associated with the whole model in a sensitivity analysis is not only a best practice, but may also help identify a broad range of influential parameters that can be manipulated in a management context. For example, while it is not possible to directly manipulate dispersal ability of a species *per se* as it is an intrinsic life-history trait, dispersal rates may be influenced by managing habitat features, such as the size of habitat patches and their configuration across the landscape. The potential influence of such factors are only likely to be revealed if these landscape-level habitat attributes are systematically varied in addition to demographic parameters in a sensitivity analysis (e.g., *Naujokaitis-Lewis et al.*, *2013*). Given that coupled SDM-population dynamics models are used to inform species and habitat recovery planning (*Camaclang et al.*, *2014*) exploring the influence of habitat structure in addition to demographic rates in a sensitivity analysis may reveal insights into alternative factors that may be responsive to management actions.

In this paper we investigate the relative influence of uncertainty associated with habitat and demographic parameters on model predictions for an endangered plant species by way of a global sensitivity analysis (GSA). In a GSA the values of multiple parameters are varied concurrently across their full parameter space, which represents their entire range of uncertainties and can account for interactions among parameters (*Saltelli & Annoni*, *2010*). To achieve this objective, we developed GRIP 2.0, a novel decision-support tool that automates GSAs of coupled SDM-population dynamics models. GRIP 2.0 extends the functionality of an earlier version of this tool, GRIP 1.0 (*Curtis & Naujokaitis-Lewis*, *2008*). In addition to evaluating the relative influence of uncertainty in demographic parameters associated with the population dynamics model, GRIP 2.0 addresses uncertainty associated with the baseline habitat maps (i.e, SDM predictive output). First, GRIP 2.0 can evaluate SDM-based uncertainties associated with habitat

maps derived from applying ensemble approaches whereby a number of alternative models (e.g., based on different model algorithms) are fit and projected to explore a range of outcomes (*Araújo & New*, *2007*). Second, if a single SDM was specified, the tool can vary spatially-explicit habitat attributes including habitat suitability values and suitability threshold values to convert maps to binary habitat vs non-habitat. Here, users can specify the degree of spatial-autocorrelation among habitat suitability values. In both cases, it is possible to vary the size and number of habitat patches in the GSA. These two approaches to varying SDM-based uncertainties and habitat patches enable GRIP 2.0 applications to address both macroecological and landscape-scale questions. By targeting multiple types of uncertainty associated with model structure, parameters, and natural variation and stochasticity connected to SDMs and population dynamics models, we capture dominant sources of variation in coupled models (*Naujokaitis-Lewis et al.*, *2009*). We applied GRIP 2.0 to a published coupled model of the threatened plant species, whitebark pine (*Pinus albicaulis*) (*Ettl & Cottone*, *2004*), as a proof of concept to address our objective and to evaluate the application of this decision-support tool.

## METHODS

### Whitebark pine coupled SDM-population dynamics model

Whitebark pine is classified as an Endangered species (*Mahalovich & Stritch*, *2013*) in decline primarily due to blister rust infections (*Cronartium ribicola* J.C. Fisch.), although the mountain pine beetle (*Dendroctonus ponderosae*) epidemic and fire suppression are also major threats. The original stochastic spatially-explicit metapopulation dynamics model was undertaken to evaluate the influence of blister rust infection on whitebark pine viability and persistence in Mount Rainier National Park, Washington, USA, where it is in danger of local extinction (*Ettl & Cottone*, *2004*). As such, it is also referred to as a population viability analysis (PVA).

The whitebark pine model (*Ettl & Cottone*, *2004*) was implemented using RAMAS GIS software (*Akçakaya*, *2002*) and relied on the RAMAS Spatial module to identify populations (i.e., habitat patches) using a raster-based habitat suitability map of a 420.3 km$^2$ region in the park. This population map provides the basis on which metapopulation dynamics are simulated in the RAMAS Metapop module. A total of 46 populations were identified by the RAMAS patch detection algorithm, which was consistent with the actual distribution of trees (*Ettl & Cottone*, *2004*). The authors used aerial photography to estimate initial abundances within each of the populations. Vegetation plots were used to estimate the proportion of individuals in each of the 13 stage classes and the final stage matrix included 24 transition probabilities that modeled both healthy and trees infected with blister rust. The stage classes included four seedling stages, saplings, infected saplings, non-reproductive adults, infected non-reproductive adults, healthy adult trees (Class 1), and three adult stages with various degrees of blister rust infection (Classes 2–4; Table 1). Adults in Class 1 through 4, non-reproductive, and infected non-reproductive adults were affected by a ceiling model of density dependence.

Seed dispersal among pairs of populations was distance-dependent whereby dispersal rates are calculated as a function of distance between populations and the parameters of

**Table 1   Input parameters, sampling distributions and parameter ranges, and brief description of all factors varied in the global sensitivity analysis of the whitepark pine metapopulation population viability analysis.** RAMAS module refers to the specific sub-program where the parameter is specified. Parameters varied in RAMAS Spatial are unique to GRIP 2.0, whereas those varied in RAMAS Metapop were introduced in GRIP 1.0 (*Curtis & Naujokaitis-Lewis, 2008*). Parameters with RAMAS Spatial/Metapop specified include habitat-specific environment-demography relationships varied in both modules.

| Input factor[a] | Distribution and sampling range | Description | RAMAS module |
|---|---|---|---|
| Habitat suitability (HS) map | GRIP 2 has 4 options for varying habitat suitability values: <br><br> 1. Random normal: $N$(mean = mean of HS values in original landscape, SD = standard deviation of HS values in original landscape) <br><br> 2. Spatially-autocorrelated: HS surface derived from a simulated gradient where the degree of spatial autocorrelation between cell values can be modified. Uses functions from the 'randomFields' R package <br><br> 3. Ensemble: Uses ensemble predictions and the measure of uncertainty to vary new HS values. Ensembles could be based on multiple types of SDM algorithms used to model species distributions (e.g., GAM, GLM, RF, BRT). The current implementation of GRIP2 resamples HS values using a random normal variate with the mean based on the ensemble prediction for that gird cell with a SD based on the uncertainty estimate from the ensemble model (i.e., the model-based measure of variation) for that grid cell. <br><br> 4. Not varied | Spatially explicit habitat suitability values (i.e. raster files). These correspond to the predictive outputs of species distribution models. HS values are rescaled where the default settings are: the minimum is the HS threshold and the maximum is the highest HS value of the original landscape. Options exist to specify a theoretical maximum. | RAMAS Spatial |
| Neighborhood distance | $N$(Original value, 10% CV) | Used to find distinct habitat patches, represents the spatial scale at which the population can be assumed to be panmictic | RAMAS Spatial |
| Distance measure among habitat patches | $D$('edge to edge', 'center to edge', 'center to center') | Measure used to calculate the distance among pairs of patches, edge and center refer to the location on the patch where the measure starts or ends | RAMAS Spatial |
| Habitat suitability threshold | $N$(original value, 10% CV) | Habitat suitability value used as the threshold to distinguish between non-suitable and suitable habitat on the raster habitat suitability map. Any grid cell value above the threshold will be considered for inclusion as a population (i.e., habitat patch) | RAMAS Spatial |
| Number of patches | $N$(original number, 50% CV) | | RAMAS Spatial |
| Initial abundance | $N$(mean value per patch is a function of total habitat suitability, CV = 10%) | | RAMAS Spatial/Metapop |
| Carrying capacity | $N$(mean value per patch is a function of total habitat suitability, CV=10%) | | RAMAS Spatial/Metapop |
| Rmax | $N$(original value, 10% CV) | Maximum growth rate | RAMAS Metapop |
| Catastrophe extent | $D$(local, regional) | Randomly varies spatial extent of catastrophe | RAMAS Metapop |

**Table 1** (*continued*)

| Input factor[a] | Distribution and sampling range | Description | RAMAS module |
|---|---|---|---|
| Catastrophe probability | $N$(original value, 10% CV) | Probability of catastrophe occurring | RAMAS Metapop |
| Catastrophe intensity | $N$(original value, 10% CV) | Magnitude of catastrophe effect | RAMAS Metapop |
| Dispersal survival | $U(0, 1)$ | Proportion of dispersers that live | RAMAS Metapop |
| Dispersal rate | $N(0, 0.1)^*$ dispersal rate | Each dispersal rate is varied by a constant value | RAMAS Metapop |
| Number of connections | $U(0$, number of pairwise population connections possible) | Varies number of population pairs connected through dispersal | RAMAS Metapop |
| Among-population correlation coefficient of vital rates | $N(0, 0.1)^*$ correlation coefficient | Varies magnitude of correlations in vital rates among population pairs | RAMAS Metapop |
| Seed survival | $L$(original value, original value) | Seed stage | RAMAS Metapop |
| Seedling 1 survival | $L$(original value, original value) | 1 year old seedling | RAMAS Metapop |
| Seedling 2 survival | $L$(original value, original value) | 2 year old seedling | RAMAS Metapop |
| Seedling 3 survival | $L$(original value, original value) | 3 year old seedling | RAMAS Metapop |
| Seedling 4 survival | $L$(original value, original value) | 4 year old seedling | RAMAS Metapop |
| Sapling mortality | $L$(original value, original value) | Sapling | RAMAS Metapop |
| Infected sapling survival | $L$(original value, original value) | Infected sapling | RAMAS Metapop |
| Nr adult survival[b] | $L$(original value, original value) | Non-reproductive adult | RAMAS Metapop |
| Infected $n$ survival[b] | $L$(original value, original value) | Infected non-reproductive adult | RAMAS Metapop |
| Class 1 fecundity and survival[b] | $L$(original value, original value) | Healthy adult trees | RAMAS Metapop |
| Class 2 fecundity and survival[b] | $L$(original value, original value) | Branch infected adult tree | RAMAS Metapop |
| Class 3 fecundity and survival[b] | $L$(original value, original value) | Bole infected adult tree | RAMAS Metapop |
| Class 4 fecundity and survival[b] | $L$(original value, original value) | 50% crown loss infected adult tree | RAMAS Metapop |

**Notes.**

[a]Selected distributions and their parameters: $D$ = discrete distribution (discrete value$_1$, discrete value$_x$), where each value has equal probability of selection; $N$ = normal distribution (mean, standard deviation—sometimes expressed in terms of coefficient of variation, % CV); $L$ = lognormal distribution (mean, standard deviation); $U$ = uniform distribution (minimum, maximum).

[b]Denotes that the stage was included in the model of density dependence.

the dispersal-distance function of the original model. The general form of the model is:

$$m_{ij} = a \times \exp\left(-D_{ij}^c / b\right)$$

where $m_{ij}$ is the dispersal rate between the $i$th and $j$th populations, $a$ (scaling parameter), $b$, and $c$ are function parameters, and $D_{ij}$ is the distance between a pair of populations. The dispersal distance function parameters for the whitebark pine model were $a = 0.1$, $b = 0.5$, $c = 1$, and $D_{\max} = 2.5$ km (maximum dispersal distance).

Blister rust disease was modeled as a catastrophe where the probability of invasion of a site corresponded to the average invasion time of 8 years. When blister rust occurs, healthy individuals become infected, leading to declines in survival and fecundity rates.

Specifically, saplings, non-reproductive adults, and healthy adults (Class 1) transition to infected sapling, infected non-reproductive adults, and Class 2 adult stage, respectively. Once sapling or non-reproductive adult stages become infected, they continue on to death, while reproductive adult stages cycle through to Class 4, where fecundities are lower. The proportion of individuals that transition into an infected class was based on empirical data and calibration tests in the original model. The model includes both demographic and environmental stochasticity, with the latter based on a lognormal distribution. A full description of the model is detailed in *Ettl & Cottone* (*2004*) and stage classes are defined in Table 1.

RAMAS Spatial uses spatial data representing habitat requirements for a species, a user-defined habitat suitability function, and habitat-specific demographic relationships to derive metapopulation structure (*Akçakaya & Root*, *2005*). The original study included a habitat suitability map at a resolution of 0.02 km with suitability values ranging from 0 to 5 and a threshold value of 1 was applied to distinguish between non-suitable and suitable habitat. We estimated habitat-specific demographic relationships because these were not provided by the authors. Using data available in the original whitebark pine PVA, we fit linear regression models to define the relationships of total habitat suitability (*ths*, sum of habitat suitability values of all raster cells belonging to a patch), to patch-specific initial abundances and carrying capacity, separately. We used *ths* instead of habitat area as this incorporates a measure of habitat area weighted by habitat quality. Our estimated relationship between *ths* and initial abundance was;

$$\text{Initial abundance} = 227.32 * ths$$

with an $r^2 = 0.780$, $p < 0.01$. The relationship between *ths* and carrying capacity, $K$ was;

$$K = 0.8449 * ths$$

with an $r^2 = 0.765$, $p < 0.01$. Apart from standardizing the time horizon to 100 years and extinction threshold to 0, and formatting the input files for use with GRIP 2.0 (see Appendix S1) the model was not further modified. We assumed that the remaining parameter values and model structure synthesized the best available information.

## Overview of GRIP 2.0

We created GRIP 2.0 in R (*Generating Random Input Parameters*; developed and tested in v. 2.7.1 through 3.2.1 (*R Core Team*, *2015*), which interacts with RAMAS Spatial and Metapop, two modules of the software RAMAS GIS (*Akçakaya & Root*, *2005*), and uses R-spatial packages to facilitate spatial and geostatistical analyses. There are two versions available, each compatible with a one of RAMAS GIS v 5.0 or v 6.0. GRIP 2.0 builds on GRIP 1.0 (*Curtis & Naujokaitis-Lewis*, *2008*) and uses Monte Carlo methods to vary parameters (1) directly on the grid-based habitat suitability map used as an input to RAMAS Spatial, (2) specified in the RAMAS Spatial module, and (3) specified in the RAMAS Metapop module (Fig. 1). Unique sets of input parameters for the Spatial and Metapop modules of RAMAS GIS are drawn from user-defined random distributions reflecting parameter uncertainty and the range of plausible values.

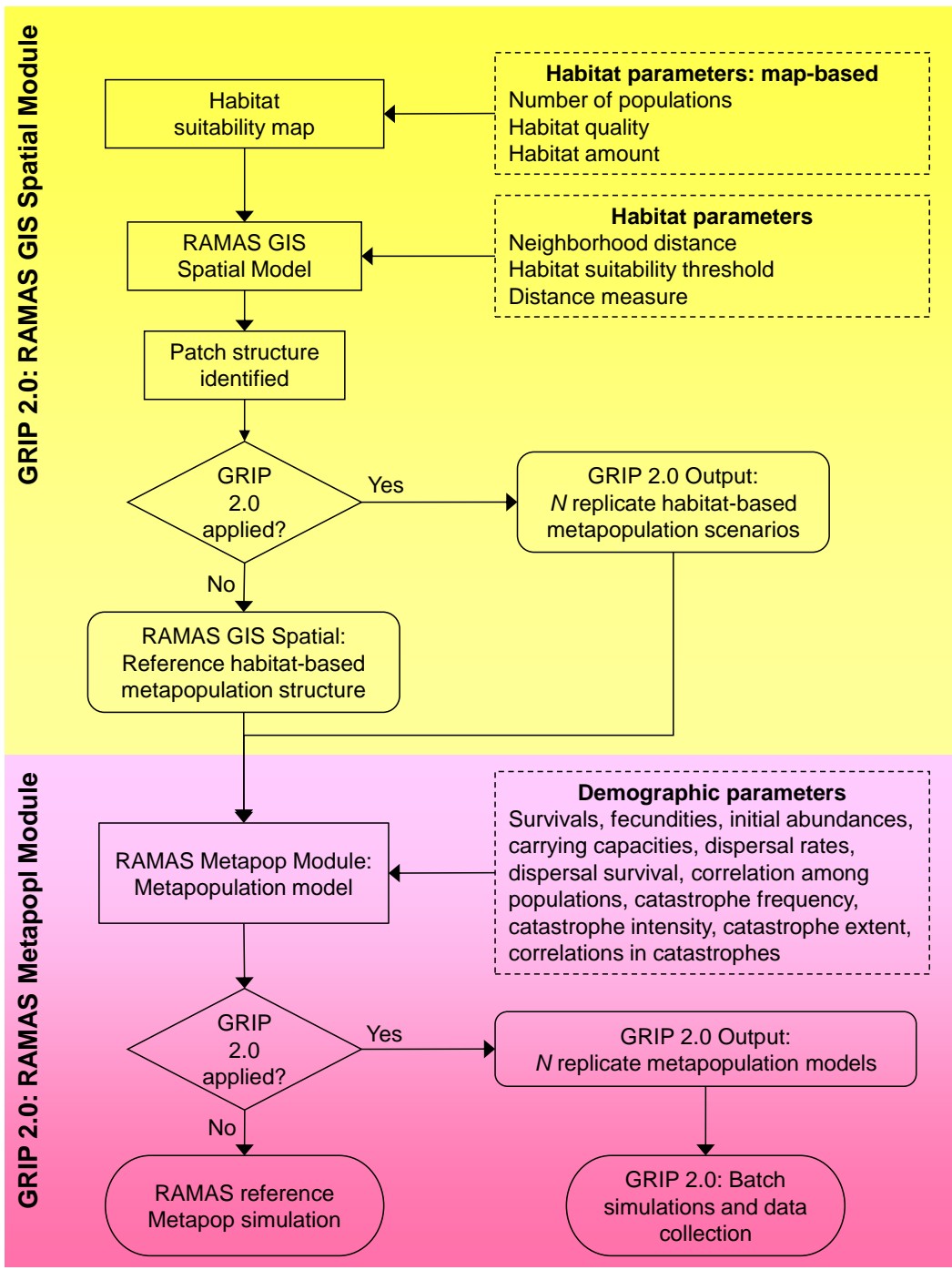

**Figure 1** **Flowchart of the global sensitivity analysis program GRIP 2.0.** GRIP 2.0 implements and automates global sensitivity analyses of coupled SDM-population dynamics models, created using RAMAS GIS, for comparing the relative influence of demographic parameters and habitat attributes on predicted extinction risk. Parameters varied by GRIP 2.0 are indicated by a dashed rectangle, model steps by a rectangle, and outputs by a rounded rectangle.

Once the replicate landscapes and corresponding metapopulation models are defined, GRIP 2.0 runs the replicate simulations in batch mode, and collates the input parameter values and predictions into a comma delimited file for subsequent analyses. Thus, GRIP 2.0 automates the generation of unique sets of replicate stochastic simulation files and manages the simulations with batch files, which would otherwise be prohibitive if undertaken manually (*McCarthy, Burgman & Ferson*, *1995*). The code of this global sensitivity analysis program is annotated and easily customized to reflect a particular species' biology or address related research questions. In the following sections, we briefly describe how habitat attributes and demographic parameters are varied in GRIP 2.0, but details are also annotated in Appendix S1 and the landscape generator and routine for varying demographic parameters is described in more detail in Appendix S2.

## Generation of alternative landscapes

GRIP 2.0 generates alternative landscapes by varying the number of patches (i.e., populations; herein we use the terms patch and population interchangeably), patch size, and the habitat suitability value associated with each raster spatial data cell directly on the original habitat suitability and patch maps (Table 1). The current version of GRIP 2.0 includes four options for varying habitat suitability maps: 'random.normal', 'spatially autocorrelated', 'ensemble', and 'no variation' (Table 1). In the 'random.normal' option, habitat suitability (HS) values are simulated by drawing a value for each grid cell from a normal distribution based on the mean and standard deviation of HS values within the reference landscape. There is no spatial autocorrelation amongst habitat suitability values. The 'spatially autocorrelated' option provides flexibility to vary the degree of autocorrelation among HS values simulated using the RandomFields R package (*Schlather et al.*, *2015*). Base settings include a Gaussian model of spatial autocorrelation with a mean of 0, variance of 5, nugget value of 1, and a scale of 10, creating a generally highly correlated surface. All HS values are set to be equal to or greater than the newly sampled HS threshold value (see below). The 'ensemble' approach can evaluate SDM-based uncertainties associated with alternative models (e.g., based on different model algorithms) whereby a number of models are fit (an ensemble) and then combined into a consensus prediction. In GRIP 2.0, users must include a raster-based ensemble SDM prediction layer reflecting a consensus estimate of habitat suitability and a raster reflecting uncertainties in that prediction. The sensitivity analysis of habitat suitability thus reflects cell-based uncertainty in the prediction and captures error propagation across multiple SDMs. Finally, the 'no variation' option ensures that HS is not included in the GSA. Given information contained in the original model, we varied HS for the whitebark pine based on the 'random.normal' option.

Additionally, two user-specified settings in RAMAS Spatial that inform its patch detection algorithm were varied: the habitat suitability threshold, which is used to distinguish between unsuitable and suitable cells, and the neighbourhood distance value, used to identify spatially discrete patches of suitable habitat. For further details see Table 1.

## Variation of demographic parameters

GRIP 2.0 generates unique replicate metapopulation models (one for each replicate landscape model) by varying parameters specified in RAMAS Metapop. A total of 23

demographic parameters specified in the whitebark pine metapopulation model were varied in the GSA (Table 1). These included stage-specific survival and fecundity rates, dispersal rates, dispersal survival, correlation of vital rates among populations, and catastrophe parameters. All demographic parameters were varied using sampling distributions and parameter ranges as specified in GRIP 1.0, a version of this freeware developed for spatial PVAs that are not based on spatially-explicit maps of habitat or habitat suitability (*Curtis & Naujokaitis-Lewis*, *2008*). We did not vary the model of density dependence to retain the original model structure to the extent possible. Further details governing the sampling distributions, mean and measure of variation for each parameter varied in the GSA are described in Table 1 and Appendix S2. As with other parameters not varied in this version of GRIP 2.0, the code is extensively annotated and customizable. To the extent possible, we used information from the original model to select biologically relevant uncertainty estimates, and in the case where these were not specified we either applied a 10% coefficient of variation or sampled from a uniform distribution (Table 1). While this 10% value is somewhat arbitrary, we used the best available information to select realistic uncertainty estimates. Users are able to modify these ranges, and even sampling distributions to better reflect the research and management context of their study system.

## Simulations and data analysis

Using the whitebark pine map as the original reference landscape, we created a total of 10,000 replicates, where refers to the total number of final model configurations that includes variation in landscape/habitat suitability parameters (Fig. 2), as well as demographic parameters. Each of the replicate metapopulation dynamics models consisted of 1,000 stochastic runs using a 100 year time period. For each replicate landscape file, we calculated landscape composition and configuration metrics using RAMAS GIS. The landscape metrics included number of patches, total amount of suitable habitat (an integrated measure of both habitat amount and suitability), mean patch area, edge to area ratio, and connectivity, calculated as the proportion of all population pairs that are linked through dispersal. A full description of the landscape metrics calculated by RAMAS GIS and associated mathematical formulae is available in *Akçakaya & Root* (*2005*).

We applied a boosted regression tree (BRT) to rank the relative influence of habitat-based measures of landscape pattern and demographic parameters on the binary response variable, conservation status. Conservation status was calculated based on the probability of extinction over a 100-year time period with 0.1 defined as the threshold for distinguishing metapopulations expected to be not at risk from those expected to be at risk of extinction. This benchmark corresponds to international criterion for listing species as Vulnerable (Criterion E; *IUCN*, *2001*). We used the machine-learning method of BRTs based on our interest in understanding the relative importance of different parameter uncertainties on model outcomes and additional flexibilities of BRT for our analysis purposes (*De'ath*, *2007*; *Elith et al.*, *2008*). GRIP 2.0 provides the means to generate and propagate uncertainties while decisions regarding how to analyze such outcomes are flexible, with BRT representing one option. The BRT functionality was not coded into GRIP 2.0 but was a supplemental

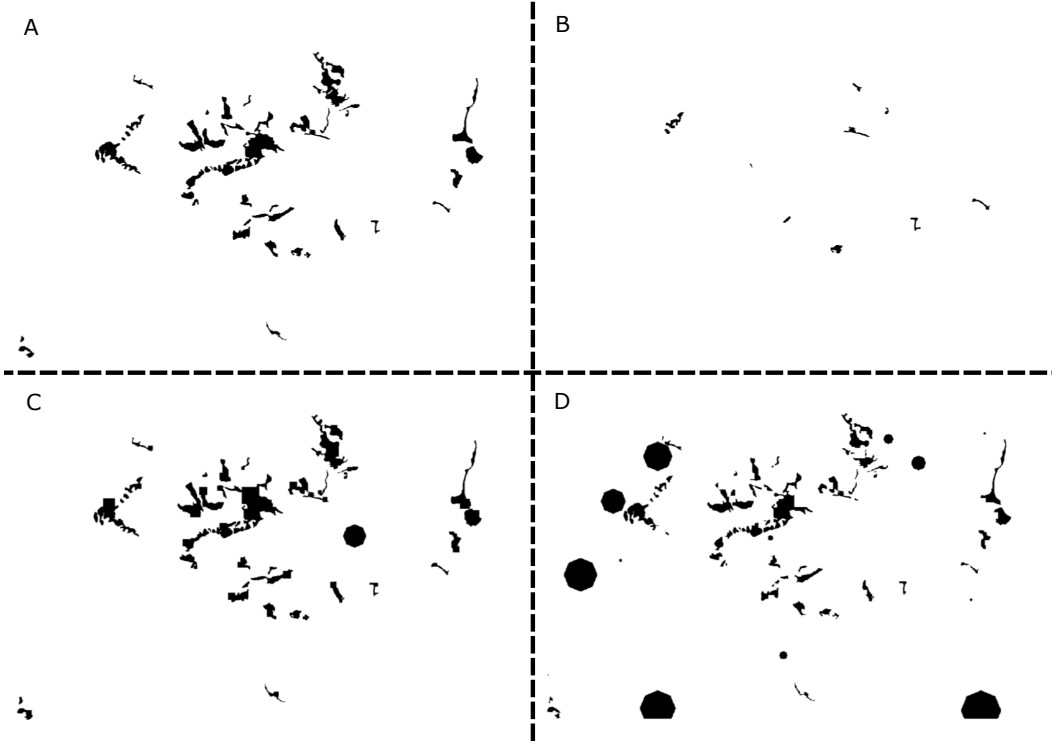

**Figure 2 Examples of simulated landscapes based on the original whitebark pine model.** The original 25.32 km × 16.6 km landscape map for whitebark pine (modified from *Ettl & Cottone, 2004*) includes 46 patches (A), and three of the simulated landscape maps (B–D) created using GRIP 2.0 includes 10, 43, and 83 patches, respectively. Of the original patches remaining in landscapes B and D, all patches have decreased in size, while the original patches remaining in landscape C have increased. Overall, the extent of the landscape for each replicate simulation remains constant but composition, configuration, and habitat suitability values have changed. For ease of representation, the landscape is depicted in a binary format where suitable habitat is black and unsuitable habitat is white. New patches created by the GRIP 2.0 landscape generator are assumed to be approximately circular in shape to take advantage of existing functionality of R-spatial packages.

statistical analysis performed once all replicate simulations were created and run using GRIP 2.0 and RAMAS.

We specified a binomial error structure and link function and used untransformed data in the BRT as it does not require data transformations (*Elith et al., 2008*) . We applied a tree complexity (*tc*) value of 2, which fits the BRT with up to two-way interactions. Learning rate (*lr*), which determines the contribution of each tree as it is added to the model, was specified at a value of 0.01, which was optimized to ensure a minimum of 1,000 trees were fit for the model (*Elith et al., 2008*). Prediction accuracy was measured using the percent deviance explained by the model, which measures the goodness of fit between the prediction and observed values. The relative influence of each predictor variable was assessed by calculating its contribution to reducing the overall model deviance of the BRT model. We identified important modeled interactions by quantifying the strength of pairwise interactions while keeping non-focal variables at their mean values. The BRT model, including relative influence of predictors, and evaluation and visualization of

**Table 2** Relative contribution of the ten most important variables to extinction status of the whitebark pine based on the 2-way interaction boosted regression tree.

| Variable | Relative contribution (%) | Type of variable |
|---|---|---|
| Total habitat amount | 40.3 | Habitat |
| Survival class 4 | 13.7 | Demographic |
| Survival class 3 | 12.8 | Demographic |
| Catastrophe intensity | 8.9 | Demographic |
| Mean carrying capacity | 5.9 | Habitat |
| Mean correlations | 2.6 | Demographic |
| Mean habitat suitability | 1.8 | Habitat |
| Mean dispersal rate | 1.5 | Demographic |
| Fecundity class 1 | 1.2 | Demographic |
| No. of populations | 1.1 | Habitat |

two-way interactions were performed in R (*R Core Team*, *2015*) using the 'gbm' package (*Ridgeway*, *2015*) and functions available in *Elith et al.* (*2008*).

# RESULTS

## Influential parameters on whitebark pine conservation status

The baseline model of the whitebark pine in the presence of blister rust predicts a dramatic decline of the metapopulation in Mt. Rainier National Park (*Ettl & Cottone*, *2004*). The two-way interaction BRT model resulted in a high level of explanatory power. The model accounted for 90% of the mean total deviance (1 − mean residual deviance/mean total deviance; 1 − 0.014/0.137). For the 2-way interaction BRT model, the loss function was minimized at 4,250 trees, and the model was optimized with a learning rate of 0.01 and a bag fraction of 0.05. The five most important variables based on the BRT included total habitat amount (40.3%), survival class 4 (13.7%), survival class 3a (12.8%), catastrophe intensity (8.9%), and mean carrying capacity per patch (6.6%) (Table 2).

The partial dependence plots in Fig. 3 illustrate the relationship between the four most important variables and conservation status after accounting for the average effect of all other variables. These plots suggest that a conservation status of 'Vulnerable' is a function of low amounts of total habitat in the landscape, low survival rates of trees in classes 3a and 4, and stronger negative effects of catastrophes (i.e., blister rust).

## Importance of interactions

The BRT analysis identified a number of strong interactions influencing whitebark pine conservation status. The strongest interaction was between total habitat amount and survival rate of trees in class 4 (Fig. 4A). Specifically, extinction risk increases more rapidly for lower values of survival and lower amounts of habitat in the landscape, but as survival increases above 0.8, less amount of habitat is required for the conservation status to remain 'not-at-risk'. Based on the second most important interaction, the influence of catastrophe intensity on extinction probability is conditional on the total amount of habitat in the landscape (Fig. 4B). In other words, when the effect of catastrophes is weaker, less

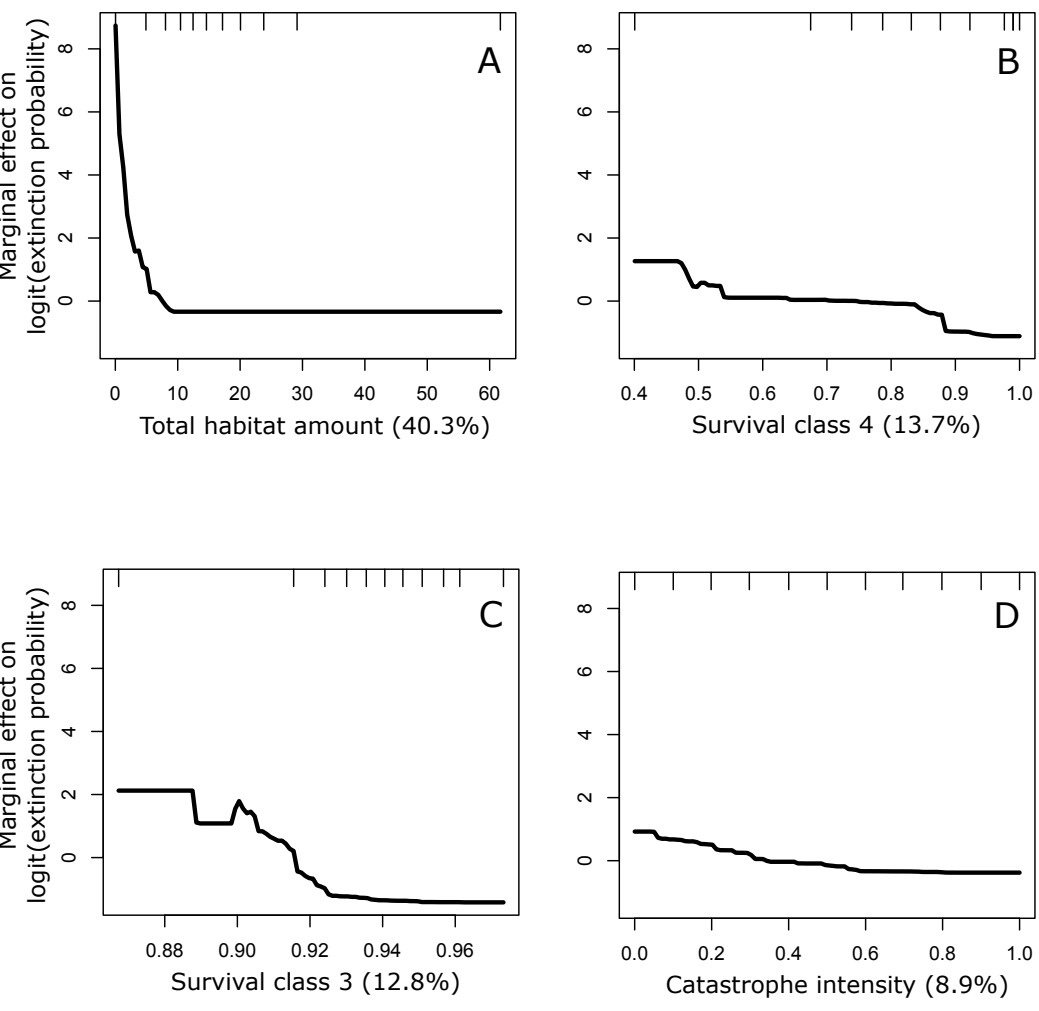

**Figure 3** **Partial dependence plots for the four most important predictors of conservation status of the whitebark pine.** The four predictors include: (A) total habitat area, (B) survival class 4, (C) survival class 3a, and (D) catastrophe intensity. Importance was ranked based on each predictors' contribution to reducing the overall model deviance (value in parentheses expressed as a %). Conservation status was calculated based on the probability of extinction over a 100-year time period where values $\geq 0.1$ were considered to be at risk of extinction (status: 1) and values $<0.1$ were considered not at risk (status: 0). This benchmark corresponds to international criteria for listing species as 'Vulnerable'.

amount of habitat is needed for the predicted status to remain not-at-risk whereas when catastrophes have a larger effect more habitat is needed to buffer the risk status rank. Both of these interactions demonstrate the importance of habitat amount in mediating the negative effects of blister rust on whitebark pine extinction risk within this landscape.

## Habitat thresholds

Based on qualitative assessments, the 3-dimensional plots revealed thresholds in habitat-based features. For the two strongest interactions (Fig. 4), across all survival rates and catastrophe intensity values, below total habitat amounts of 2 km$^2$ the predicted risk of extinction increases more steeply. Despite the strong interaction among these pairs

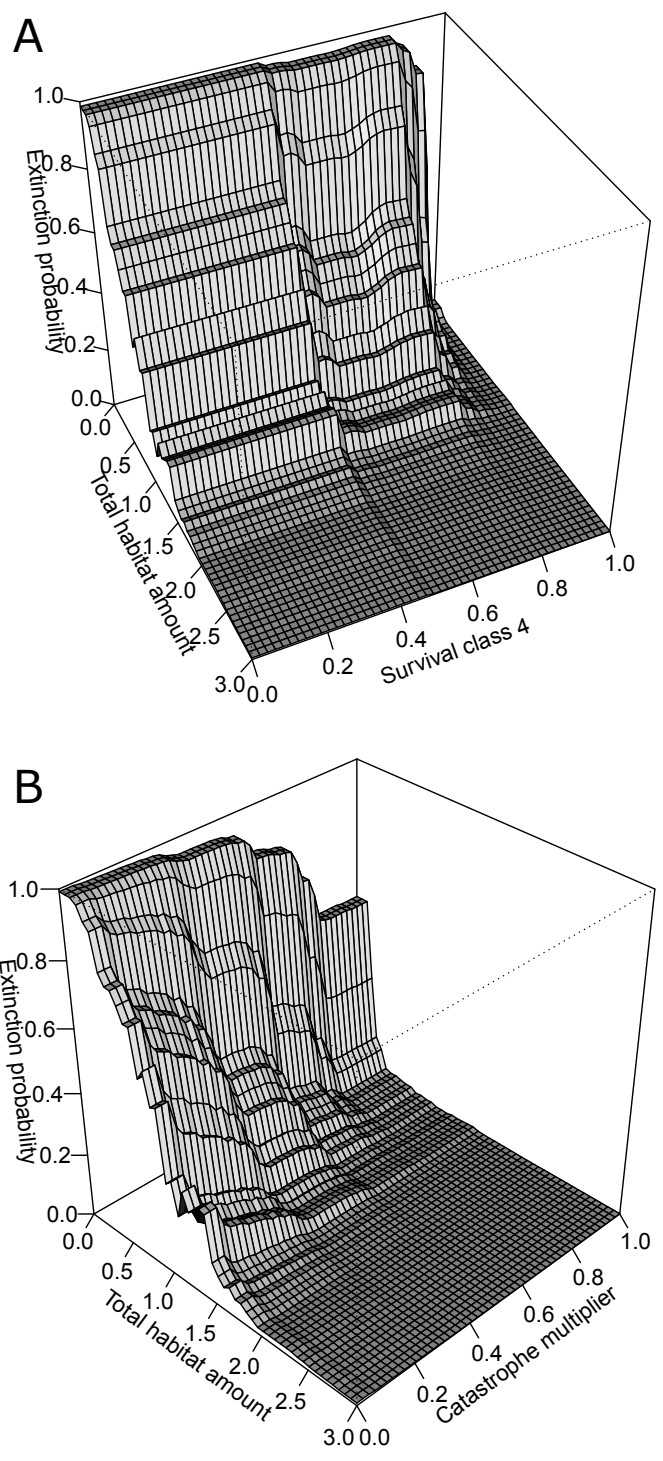

**Figure 4  Three-dimensional partial dependence plots for the two strongest interactions based on a global sensitivity analysis of the whitebark pine metapopulation model.** All other variables not plotted remain at their mean value. (A) Interaction between total habitat amount and survival class 4, and (B) interaction between total habitat amount and catastrophe intensity.

of variables, this suggests the existence of thresholds in total habitat amount for the Mount Rainier whitebark pine metapopulation infected with blister rust. As the number of populations did not interact strongly with other demographic parameters or habitat features, the partial dependence plots may be interpreted directly: thresholds were evident for the number of populations where conserving >20 populations resulted in predicted conservation status of 'not-at-risk', based on international listing criteria.

## DISCUSSION

The use of coupled SDM-population dynamics models that relate land-use and climate dynamics to population-level demographic parameters present large potential applications for endangered species recovery and habitat planning. However, the potential for the propagation of errors and uncertainties throughout the modeling process represents a true concern. We developed a novel tool to illustrate the importance of advancing the application of global sensitivity analyses to targeting each component model of coupled SDM-population models. Based on the whitebark pine model, the most important variables on predictive outcomes included a combination of demographic and landscape habitat features. The inclusion of habitat features in a sensitivity analysis broadens the scope of potentially effective management actions aimed at supporting persistence of endangered species beyond the usual focus on vital rates. Such assessments are fundamental not only to help identify parameters requiring further data collection to reduce model uncertainty but also for evaluating and prioritizing costly habitat-based management recommendations for endangered species.

### Simulating alternative realizations of landscape and habitat structure

Landscape level experiments involving adequate replication are often difficult to implement due to issues of scale. Consequently, experimentation by simulation using landscape generators presents a viable alternative to develop a better understanding of the relationship between landscape pattern and process (*Gardner & Urban*, *2007*), predict species response to landscape change (*Tischendorf*, *2001*), and more generally to develop and test hypotheses. Because GRIP 2.0 modifies an existing reference landscape, certain landscape elements, such as size and shape of some patches, are retained in replicate simulations. This leads to more realistic simulated landscapes, unlike neutral or multi-fractal models (*With & Crist*, *1995*). Although not applied to the whitebark pine model, GRIP 2.0 allows users to include a landscape mask, enabling habitat creation only in those regions outside of the mask. This aspect increases the functionality GRIP 2.0 and the realism of its outputs; a mask may contain intractable barriers to habitat creation, such as roads, representing landscape constraints that many analysts and managers must contend with.

Replicate landscapes produced by GRIP 2.0 represented a wide range of landscape structural variation allowing an evaluation of the influence of a broad range of structural attributes on extinction probability of whitebark pine, and by the same token, a broad range of scenarios to explore for conservation and management planning. By using Monte Carlo simulations, landscape features modified included the amount of habitat, patch sizes, or the degree and directionality of spatial autocorrelation in replicate landscapes.

Such flexibility is important given simulation studies and our results that have identified thresholds related to levels of habitat within a landscape below which species viability rapidly declines (*Swift & Hannon*, *2010*). The GRIP 2.0 code is highly annotated allowing users to understand, scrutinize, and modify the code to reflect a particular species' biology, sampling distributions, or landscape dynamics.

The original whitebark pine model did not model consequences of climate change on future projections. Thus our implementation of GRIP 2.0 to the whitebark pine model does not address uncertainty in spatio-temporal projections of habitat suitability under future climate change associated with selection of General Circulation Models, for example. However, this is not an inherent limitation of the tool as GRIP 2.0 can be modified to integrate this additional source of uncertainty as in *Naujokaitis-Lewis et al.* (*2013*). Additionally, while we did not explicitly address uncertainty in the habitat-demographic relationship, our GSA approach implicitly addresses this source of variation by varying population-specific initial abundances and carrying capacities. Should models incorporate temporal trends in vital rates associated with estimated habitat-demographic relationships, GRIP 2.0 is customizable to reflect this potential source of variation.

## The relative influence of parameters and their interactions

By varying multiple parameters simultaneously using probability distribution functions to represent the plausible range of parameter values, GRIP 2.0 performs a global sensitivity analysis, in which parameters are varied concurrently over the plausible range of parameter space (*Saltelli et al.*, *2006*). Varying habitat features associated with habitat suitability maps, such as habitat amount, habitat suitability, and number of patches enabled a comparison of the relative influence of habitat-based attributes on whitebark pine meta-population dynamics relative to demographic parameters. Outcomes of our global sensitivity analyses indicated that both demographic and habitat factors influence predictions of whitebark pine persistence. Our GSA identified different influential parameters from the original study, which did not vary habitat-based factors (*Ettl & Cottone*, *2004*). Based on the predicted trajectories of individual whitebark pine subpopulations over time, *Ettl & Cottone* (*2004*) concluded that size and distribution of subpopulations influenced the trajectory of individual populations. The authors also concluded that the model was more sensitive to changes in the vital rates of healthy trees than infected trees, and less sensitive to changes in the vital rates of mature trees than of younger trees, and more sensitive to the frequency of invasion than by the effect of blister rust on vital rates. Without applying a GSA, however, the authors were not able to rank the relative influences of these factors on the dynamics of the meta-population as a whole, or identify key interactions among parameters and thresholds that could be used to inform management decisions. By contrast, the results of our GSA indicated that model predictions were more sensitive to older stage classes that were also infected by blister rust disease. We also identified certain habitat variables as highly influential, which broadens the range of information that can be used to develop effective management actions. Our simulations were standardized to 100 years (as per criterion E for IUCN species assessments), but with poor recruitment/low survival, this metapopulation may be at greater risk over the longer term (*Ettl & Cottone*, *2004*) .

The relatively high rank of certain habitat factors, such as total amount of habitat, highlights the importance of spatial habitat-based landscape features as important drivers of whitebark pine persistence as a way to mediate the negative consequences of blister rust disease. Spread of the fungal blister rust pathogens is complex; blister rust has a five-stage life cycle and requires two hosts, whitebark pine (or any other five-needled pines) and any species in the genus *Ribes*. Predicting the spread of the blister rust through the distribution of whitebark pine individuals and populations is difficult as the pathogen spreads from *Ribes* hosts to whitebark pine via wind-borne spores, and not from tree to tree (*McDonald & Hoff*, *2001*). Although the original PVA model does not explicitly model transmission through *Ribes*, the outcomes of our GSA indicate that managing habitat parameters could help offset declines in abundance related to blister rust. Given that proximity to *Ribes* species contributes to increased rates of blister rust infection (*Smith et al.*, *2011*), future research that explicitly explores the spatial context of *Ribes*' role in disease transmission dynamics would improve our understanding of the role of multiple interacting parameters that influence whitebark pine persistence.

### Thresholds and conservation implications

Theoretical models have been used to generate hypotheses, such as the nonlinear threshold hypothesis, which predicts that species exhibit threshold responses due to the increasing influence of fragmentation below a certain amount of habitat (*Andrén*, *1994*). Although theoretical models predict that below critical threshold points, increased fragmentation results in lowered colonization success and increased extinction probability (*With & King*, *1999*), empirical validation of such responses are uncommon (*Fahrig*, *2003*). Our results corroborate the occurrence of thresholds in response to declines in both habitat amount and quality. Visual assessment of bivariate plots of probability of extinction as a function of landscape variables indicated the presence of thresholds as evidenced by strong negative exponential decay curves (results not shown). Although we did not quantitatively derive threshold points for whitebark pine, these roughly translate to 20 populations, a mean connected distance of approximately 5 km, an average habitat suitability of 2.3 per patch, and a mean patch size of $\sim 0.35$ km$^2$. Further insights into the behavior and occurrence of thresholds in real settings may help identify conservation strategies that specify minimum patch size targets, or provide guidance related to landscape level measures, such as habitat amount (*Betts, Forbes & Diamond*, *2007*).

### CONCLUSIONS

Understanding the relative importance of factors influencing species extinction risk can provide information needed for the design of actions to target species recovery and persistence. Our approach provides one way to assess the role of uncertainty of multiple parameters on species persistence, and assist in the prioritization of research and evaluation of alternative management strategies (*Guisan et al.*, *2013*). In this particular case, we have shown that for a species threatened by disease invasion, both demographic and habitat-based variables rank high in terms of influence on the risk of extinction. This suggests that there are multiple options for effective management of whitebark pine, ranging

from direct actions targeting diseased individuals or mediation through habitat-based measure. However, the relative influence of such options only becomes apparent through their explicit inclusion in a global sensitivity analysis and subsequent assessment via an appropriate analysis, such as a BRT as applied in this example. Furthermore, a global sensitivity analysis framework is considered a best practice that can be incorporated into the development and communication of PVAs (*Pe'er et al.*, *2013*). Future work using coupled SDM-population dynamics models to prioritize research and evaluate management strategies would benefit from the integration of a decision-theory approach and socio-economic information to help inform investments required to achieve specified conservation objectives.

## ACKNOWLEDGEMENTS

We thank Roger Bivand and Adrian Baddeley who helped with our various programming queries, Norm Hodges and Sarah Gergel for assisting with computer support, R Akçakaya who fielded numerous technical questions related to RAMAS GIS, and Peter Arcese, Jordan Rosenfeld, and Pippa Shepherd for useful discussions. We thank the anonymous reviewers for comments on previous versions.

### Funding

J Curtis was funded by a Parks Canada Species at Risk Recovery Education and Action Fund and the Fisheries and Oceans Canada National Species at Risk Program. I Naujokaitis-Lewis was funded by a Natural Sciences and Engineering Research Council postgraduate scholarship. The funders had no role in study design, data collection and analysis, decision to publish, or preparation of the manuscript.

### Grant Disclosures

The following grant information was disclosed by the authors:
Risk Recovery Education and Action Fund.
Risk Program.
Natural Sciences and Engineering Research Council postgraduate scholarship.

### Competing Interests

The authors declare there are no competing interests.

### Author Contributions

- Ilona Naujokaitis-Lewis conceived and designed the experiments, performed the experiments, analyzed the data, contributed reagents/materials/analysis tools, wrote the paper, prepared figures and/or tables, reviewed drafts of the paper.
- Janelle M.R. Curtis conceived and designed the experiments, contributed reagents/materials/analysis tools, wrote the paper, reviewed drafts of the paper.

## Data Availability

The research in this article did not generate any raw data. We used a previously published population viability analysis model. The R code we wrote to perform our global sensitivity analysis of coupled species distribution-population dynamics model is available as a Supplemental File.

## Supplemental Information

Supplemental information for this article can be found online at http://dx.doi.org/10.7717/peerj.2204#supplemental-information.

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
