# Peer review of "Advances in global sensitivity analyses of demographic-based species distribution models to address uncertainties in dynamic landscapes"

_PeerJ, doi:10.7717/peerj.2204_

## Round 0.1 · original submission · Minor Revisions

I think this manuscript will make a valuable contribution by attracting attention to GRIP 2.0 as a tool for yielding insights into species conservation. However, the reviewers have identified various points that require clarification. In some cases, key information on methodology is missing. In order to maintain readability, you'll have to balance what methodology is included in the main text, versus what is included in an appendix. However, if the requested details can be included with minimal added text, then I suggest including them in your main text. I am also especially interested in your response to one reviewer's comment about probability distributions from an ensemble approach such as BIOMOD. Many researchers used ensemble SDMs, so it would be ideal if GRIP 2.0 can handle inputs of these probabilities. I think this point could merits specific mention in your Discussion.

In addition to the reviewer comments, I have some specific comments on the text and figures as follows:

Line 65 points b and c – I don't think these can be properly addressed with GRIP. [a reviewer made a similar comment] – place emphasis in your introduction on the types of uncertainties that are best addressed by your tool (your point a).
Line 149 Should the date be 2001?
Line 171 I think some mention of GRIP 1.0 should be made in the introduction, leading to a clear statement of why GRIP 2.0 is needed.
Line 228 Please clarify early on whether the BRT analysis is built into GRIP, or whether the BRT analysis was a supplemental analysis that you did using the GRIP output. Based on information finally given on line 258, I guess it is the latter, but it should be clarified earlier – even considering wording on line 258, one interpretation could be that GRIP calls on the gmb package to give users the BRT output (which would be convenient, but I think this is not what GRIP does).
Line 347 Earlier (Line 121) you state that your study performs as GSA, but here you state that is “approximates” a GSA. Then on 352and 410 you state “our GSA indicated” (implying actual GSA). Be consistent with wording.
Lines 400-415 It seems that many of your conclusions are based on the BRT analyses, which does not seem to be part of GRIP 2.0 (based on looking at Figure 1) . I think the conclusion should point out how GRIP and BRT work together to yield new insights, and it would also be valuable to point out any other promising approaches besides BRT that could help yield insights from the GRIP output.

Figure 2 C and D – Please briefly explain in the caption why the perfect octagon shapes appear on these maps.

Figure 4 - I think the Y axis should say “probability of extinction”, not conservation status. My understanding is that conservation status was defined as a binary variable: at risk or not at risk (line 231). I think the y-axis label on figure 3 should be similarly reconsidered, and the captions on both figures may need to be clarified. If you wanted to emphasize “conservation status” you could try using different shadings or line patterns for y-values above or below 0.1. The would need to be explained in the caption(s).

·

Basic reporting

No Comments

Experimental design

No Comments

Validity of the findings

No Comments

Additional comments

Review of ‘Advances in global sensitivity analyses of demographic-based species distribution models to address uncertainties in dynamic landscapes.’

The manuscript describes an update to a previously described tool (written in the R programming language) to facilitate the generation of RAMAS files for conducting global sensitivity analysis of PVA models that couple matrix-based population models with species distribution models. The tool generates a set of files with randomly varied parameters and the associated files for running the files as a batch and summarizing the results so that the relative impact of model parameters can be related to model outcomes in a global sensitivity analysis (GSA). The advance of this update to the tool (GRIP 2.0) is that it allows random manipulation of the underlying habitat map for inclusion of more spatial parameters in the GSA. The authors convincingly demonstrate the importance of including spatial factors in a GSA.

Overall, the manuscript is very well-written. The language is clear, concise, and easy to follow. The figures are clear and appropriate. The analysis was well constructed and described, and the conclusions are supported by the results. I only have a few minor comments and/or questions about the tool and the analysis, but nothing serious that should impede publication. Most of my comments relate more to the functionality of the GRIP tool in general rather than to the application of the tool described in the current manuscript.

1. A few additional details on GRIP ver. 1.0 and the original whitebark pine study could improve the manuscript. The reader is referred to the original whitebark pine model for the structure of the demographic model, and the reader can get a sense of it from Table 1, but a short summary describing key aspects of the previous model here might prove useful. Also, the structure of density dependence can have large effects on model outcomes, but density dependence of the model is only referenced in passing here.
2. Related to the first point, whether or not to vary a parameter in a GSA should have some biological justification (and biologically justified ranges based on either the range of uncertainty of the true value or the range of true variability). There is little to no justification presented for the parameters selected to test in the GSA.
3. When GRIP 2.0 increases the number of populations/patches, it is my understanding that a new random cell is selected to form the center of the new ‘patch’ (discrete cluster of suitable habitat). Once the new patch is assigned a size, is it possible that the new patch could be within the neighborhood distance of an existing patch and therefore subsumed into it when the RAMAS spatial module is run on the new map therefore resulting in fewer total populations than expected? If so, I don’t think it’s a problem for the functionality of the tool, but might be a point for clarification.
4. The way the tool currently varies the habitat suitability map could result in simulated landscapes that are very different from the reference landscape if there are strong spatial patterns in habitat suitability. This could result in the HS variable having more or less influence than it should in the GSA. The current method described draws a random value based on the mean and variance of the suitability over the entire reference HS map and includes spatial auto-correlation among the sampled HS values. My concern is that if there is was a something like a strong clinal pattern to the landscape that pattern would be lost when tool re-samples HS.
5. I did scan through some of the R code, but didn’t get a chance to test it. I noticed one section where there was a limitation imposed on the total number of populations based on a constraint in RAMAS. Just a helpful tip: The maximum number of populations (along with several other parameters) can be increased above the default values by modifying the RAMAS configuration file.
6. The current version of RAMAS is v. 6.0. Will GRIP 2.0 work with that version as well?
7. Not totally necessary for publication, but would it be possible to share the RAMAS files used in this analysis that are called in the accompanying R code? Or possibly a short tutorial to walk users through a worked example? I think that might be really useful for folks who might consider using GRIP 2.0.

Reviewer 2 ·

Basic reporting

No Comments

Experimental design

No Comments

Validity of the findings

No Comments

Additional comments

Naujokaitis-Lewis and Curtis introduce an updated tool GRIP 2.0 for global sensitivity analyses of coupled SDM-population dynamics models and exemplify this by re-examining an older PVA study on whitebark pine. The proposed tool is timely and important as sensitivity analysis is often neglected or insufficient in coupled SDM-population dynamics models. As a special feature over its predecessor GRIP 1.0, the new tool is now able to explicitly test for sensitivity of PVA to landscape/habitat features. The presented case study underscores the benefit of such integrated assessments that help elucidating the interactive effects of demography and habitat on population viability.

Nevertheless, I have a few concerns regarding the presentation of the tool and the case study, which I detail below. In summary, the introduction could better motivate the kinds of uncertainty explicitly analysed by GRIP 2.0, the tool could be described in more detail regarding e.g. the sampling algorithm to draw parameter values, and the design of the landscape generation tool could be better justified and the corresponding field of applications of GRIP 2.0 explicitly named.

L62-73: the authors name important sources of uncertainties but not all of these are explicitly dealt with in the ms nor in GRIP 2.0. Following this list of potential uncertainties, it should be better motivated and justified which sources are explicitly addressed. For example, the link between habitat suitability and demography is mentioned as important, but is not analysed.
L88: I certainly agree that sensitivity/uncertainty analyses are not made routinely enough in coupled SDM-population dynamics models. Nevertheless, the authors have missed some important publications of recent years that do try to consider uncertainty by local to (quasi-)global sensitivity analyses. For example, Cabral et al. 2011 (Conservation Biology 25: 73-84), Zurell et al. 2012 (Ecography 35: 590-603), Boulangeat et al. 2014 (Global Change Biology 20: 2368-2378), to name but a few.
L128-166: I’d recommend a slightly more detailed description of the model so that it is traceable without further consulting the original publication. Several model details remain obscure. For example, how were the habitat-specific demographic relationships estimated? Why was K not set to twice the initial abundance as specified in the original model? Which predictor variables were used in logistic regression to define habitat suitability? Also, a habitat suitability threshold of 1 seems unusual as logistic regression models rarely produce probability output of exactly 1. Then, several more model details are not specified at all, for example the form of density dependence (is a ceiling effect used, or are vital rates reduced if carrying capacity is reached?).
L195ff: Unfortunately, this procedure does not reflect SDM-based uncertainty as provided e.g. by ensemble methods such as biomod2 in R. Ensemble SDMs provide for each cell in the landscape a vector of probabilities that reflects local uncertainty in occurrence probability. Ideally, local habitat suitability should be sampled from such output to really capture error propagation from the different coupled modules. The choice of how habitat suitability is varied here should be justified. It is certainly motivated more from a landscape ecological perspective rather than a macroecological or global change perspective. Yet, coupled SDM-population dynamic models are used in all of these contexts and, thus, the application domain of the proposed tool should be described more explicitly, or other possibilities of representing uncertainty in suitable habitat should be discussed.
L214: please describe this ‘manner’ in more detail so it is understandable without consulting the original Grip 1.0 publication.
L218ff: Specifics on the global sensitivity analysis are missing. For example, which sampling algorithm is used to draw from the probability density functions of the different parameters – Latin hypercube sampling or similar? Does the ‘total of 10,000 replicate landscapes’ refer to the number of different model configurations within sensitivity analysis and include variation in habitat suitability as well as in demographic parameters? This is not clear and the term ‘replicate landscapes’ might be misleading here.
L235-259: The description of BRTs seem inappropriately long compared to the scarce details provided for the pine model.

Line comments:
L262-268: this rather belongs to methods.
L305/L397: there is no appendix C.
Reference list: Ettl and Cottone (2004) (see also in text, e.g. L262, L355, L369)
Figures 3+4: why is the range of habitat amount different in those two figures? (The text suggests that the terms total habitat amount and total habitat area are used interchangeably, cf. L280, L298)

---

## Round 0.2 · accepted · Accept

Great to see the expanded code with more options for users!